# Innovations in Platelet Cryopreservation: Evaluation of DMSO-Free Controlled-Rate Freezing and the Role of a Deep Eutectic Solvent as an Additional Cryoprotective Agent

**DOI:** 10.3390/ijms262010013

**Published:** 2025-10-15

**Authors:** Rahel Befekadu, Natasha Bosnjak, Michael Uhlin, Agneta Wikman, Per Sandgren

**Affiliations:** 1Clinical Immunology and Transfusion Medicine (KITM), Karolinska University Hospital, SE-14186 Huddinge, Swedennatasha.bosnjak0@gmail.com (N.B.);; 2Hematology and Regenerative Medicine (HERM), Karolinska Institutet, SE-14186 Huddinge, Sweden

**Keywords:** platelets, DMSO-free cryopreservation, deep eutectic solvents (DES), controlled-rate freezing (CRF), dimethyl sulfoxide (DMSO, Me2SO)

## Abstract

Cryopreservation is a well-established method for extending platelet shelf-life and addressing supply shortages. Traditionally, this involves dimethyl sulfoxide (DMSO) as a cryoprotective agent (CPA), but recent studies suggest that using controlled rate freezing (CRF) with only NaCl may offer a less toxic alternative. To explore further optimization, this study assessed whether adding 10% choline chloride–glycerol, a deep eutectic solvent (DES), could enhance platelet quality in CRF/NaCl cryopreservation. Ten double-dose buffy coat platelet units were divided into test (DES-treated) and control (NaCl-only) groups. After DES exposure (10% for 20 min), all units were prepared using the NaCl protocol and frozen at −80 °C with CRF equipment, then stored for over 90 days. Upon thawing and reconstitution in AB plasma, no significant differences were observed in platelet content post-thaw between control and test units (255 ± 43 vs. 257 ± 41 × 10^9^/unit), post-thaw recovery (>85%): respectively, Δψ (JC-1% pos 63 ± 15 vs. 68 ± 17), LDH (% of total 10 ± 6 vs. 9 ± 6), (CD63% 77 ± 9 vs. 82 ± 7), (CD62P % 72 ± 15 vs. 76 ± 11), (CD42b % 78 ± 9 vs. 80 ± 9), (CD61% 79 ± 9 vs. 78 ± 9), (CD41% 81 ± 11 vs. 83 ± 7), (PAC-1% 33 ± 10 vs. 32 ± 8), (Pecam-1% 78 ± 11 vs. 80 ± 8), (GPIV % 72 ± 10 vs. 74 ± 11), (LAMP-1% 26 ± 14 vs. 11 ± 9), (MPCD61+ % 41 ± 11 vs. 46 ± 10), (ROTEM CT 56 ± 7 vs. 55 ± 6), (ROTEM CFT 110 ± 70 vs. 106 ± 67) and (ROTEM MCF 35 ± 6 vs. 36 ± 6). These findings support the feasibility of CPA-free CRF-based platelet cryopreservation while maintaining functional integrity.

## 1. Introduction

Platelet cryopreservation is a critical technology in blood banking, transfusion medicine, and emergency care, enabling the long-term storage of platelets for military and clinical use [1,2,3]. Conventional methods rely on dimethyl sulfoxide (DMSO) as a cryoprotective agent (CPA). However, DMSO presents notable drawbacks, including cytotoxicity and in some cases the need for washing steps before transfusion (references), which add complexity, delay, and cost—especially in acute care and remote settings. Concerns over DMSO toxicity have therefore driven efforts to develop alternative CPAs [4,5,6,7,8,9].

A recent approach eliminates DMSO entirely, freezing and storing platelets in isotonic saline (NaCl) alone [10]. While simpler and less toxic, this method still does not sufficiently prevent freezing-induced damage, highlighting the need for optimized DMSO-free protocols with improved cryoprotection. The novel study by Ehn et al. [10], however, shows that the NaCl protocol combined with controlled-rate freezing (CRF) improves in vitro outcomes—such as post-thaw recovery and function—compared with uncontrolled freezing, although the protection hypothetically remains to be further improved. Deep eutectic solvents (DESs) have emerged as a promising class of next generation cryoprotective agents (CPAs) due to their tunable composition, low toxicity, and favorable biocompatibility. DESs are formed by combining a hydrogen-bond donor and acceptor to yield a eutectic mixture with a depressed melting point, originally conceptualized as analogs to ionic liquids. Their extensive hydrogen-bond networks confer strong solvency for biomolecules and contribute to membrane and protein stabilization—properties essential for effective cryopreservation [11,12].

While recent studies highlight the potential of systems such as proline–glycerol (Pro–Gly, 1:3), which achieve platelet recovery and post-thaw function comparable to dimethyl sulfoxide (DMSO) [11], most existing research focuses on amino acid–polyol DESs. The current study investigates a choline chloride–glycerol DES, chosen for its unique ionic and hydrogen-bonding characteristics that may provide enhanced membrane protection and reduced cytotoxicity relative to conventional and amino acid-based DESs. By broadening the range of DES components to include choline-based systems, this work expands the chemical landscape for CPA innovation and illustrates the versatility of DESs in meeting diverse preservation demands.

Choline chloride (ChCl), a naturally occurring quaternary ammonium salt, is widely used as a hydrogen bond acceptor in the formulation of deep eutectic solvents (DESs) by pairing with a variety of hydrogen bond donors (HBDs) notably creating tunable eutectic mixtures such as ChCl/urea that exhibit significantly lowered melting points, reduced flammability, and customizable viscosity and polarity [11,13]. In particular, ChCl-based DESs formed with l-ascorbic acid analogs have demonstrated viscous, glass-forming behavior along with preliminary indications of cryoprotective utility [13]. Importantly, ChCl’s intrinsic low toxicity and biocompatibility make it an attractive candidate for cryoprotective agent (CPA) development [14,15]. Despite these promising attributes, ChCl-based DESs have not yet been investigated for platelet cryopreservation, nor integrated into DMSO-free NaCl/CRF cryopreservation protocols, highlighting a clear gap in current cryobiological research, with only proline glycerol DES being evaluated in platelets to date [16].

This unexplored potential represents a valuable opportunity to advance DMSO-free platelet cryopreservation by exploiting the versatile properties of deep eutectic solvents (DES). The incorporation of DES into NaCl/CRF-based protocols may improve post-thaw platelet recovery, preserve phenotypic integrity, and enhance functional performance beyond that observed with isotonic saline alone [10]. Alternatively, results may indicate that the use of CRF with NaCl, in the absence of DMSO, already provides sufficient preservation efficacy, thereby rendering additional additives unnecessary. Such an outcome would also offer significant logistical advantages.

### Scientific Research Question

Can the addition of deep eutectic solvents (DES) further enhance the recently proposed DMSO-free NaCl protocol for platelet cryopreservation—specifically protocols using controlled-rate freezing (CRF)—by increasing post-thaw recovery, maintaining phenotype, and improving functional performance compared with isotonic saline alone? Additionally, can optimized NaCl/CRF protocols be further improved by the inclusion of novel additives such as DES to overcome remaining limitations in in vitro and potentially clinical performance?

## 2. Results

### 2.1. Platelet Covery and Integrity After Cryopreservation

All reconstituted units exhibited pH values within the expected physiological range at 37 °C (7.03 ± 0.04) and showed stable metabolite levels, confirming both the high quality of the fresh plasma and the adequacy of the reconstitution procedure following thawing. Platelets cryopreserved with the control protocol displayed a recovery of 86.9 ± 0.1%, whereas cryopreservation with the DES-based, DMSO-free method resulted in a recovery of 88.2 ± 0.1% (*p* = NS). The initial platelet content per unit prior to freezing was comparable between groups (255.4 ± 43.2 × 10^9^/unit vs. 257.0 ± 41.0 × 10^9^/unit. After thawing, control units contained 219.7 ± 28.1 × 10^9^/unit, while DES-treated units contained 225.9 ± 36.9 × 10^9^/unit (*p* = NS; Figure 1A), with corresponding MPV values of 10.1 ± 0.4 vs. 10.9 ± 0.6 (Figure 1B). Assessment of mitochondrial membrane potential (MMP) after freezing indicated pre-apoptotic changes; however, the proportion of platelets with intact MMP (JC-1-positive) remained similar between groups (63 ± 15% vs. 68 ± 17%; Figure 1C). Lactate dehydrogenase (LDH) release was low in both conditions (10.1 ± 6.1% vs. 8.8 ± 4.1% of total, Figure 1C), indicating minimal cell disintegration, consistent with the observed post-thaw recoveries.

### 2.2. Similar Phenotypic Expression After Cryopreservation

In line with evidence for spontaneous activation following thawing, the proportion of platelets expressing the activation markers CD62P (α-granule), CD63 (dense granule), and PAC-1 (binding specifically to the activated conformation of the fibrinogen receptor integrin αIIbβ3/GPIIb–IIIa) was high, but no differences were detected between groups (Figure 2A–C). Preservation of surface receptor expression after freezing and thawing is essential for platelet function. The proportions of platelets expressing GPIb (CD42b), the fibrinogen-binding integrin complex (CD61/CD41a), and the collagen receptor GPVI were well detected and comparable between groups post-thaw (Figure 2D–G). Similarly, the expression of PECAM-1, a signaling molecule involved in multiple aspects of platelet regulation, did not differ between DES and control platelets immediately after thawing (Figure 2H). Expression of the lysosomal-associated membrane protein LAMP-1, a marker of lysosomal degranulation and late-stage platelet activation, was low in both groups without any intergroup differences (Figure 2I). Taken together, these findings show that while platelets exhibited signs of spontaneous activation after thawing, no differences were identified between DES and control platelets with respect to activation markers, surface receptors, or lysosomal degranulation.

### 2.3. Heterogeneity of Microparticle Surface Phenotypes

The proportion of microparticles (MPs) within the extracellular vesicle (EV) population was consistent across study groups, with no detectable differences between the control and DES cohorts (Table 1). CD61-positive MPs constituted a stable fraction of total MPs, which is in line with previous findings and supports the concept that CD61 serves as a robust marker of platelet-derived vesicles.

In contrast, the expression patterns of TLR2, TLR4, and GPVI on MPs remain less well characterized in the literature. In our cohort, approximately half of the tested individuals demonstrated the presence of MPs positive for these receptors. The detection of TLR2-, TLR4-, and GPVI-positive MPs suggests that platelet-derived vesicles may indeed carry these receptors, thereby extending the repertoire of potential functional interactions mediated by MPs. This finding is further supported by the established expression of these structures on the platelet surface, strengthening the biological plausibility of our observations.

Taken together, the phenotypic expression analysis revealed that while CD61-positive MPs remain a consistent and expected finding across populations, the presence of TLR2-, TLR4-, and GPVI-expressing MPs points toward a broader and potentially underappreciated heterogeneity of MP surface phenotypes.

### 2.4. Viscoelastic Properties of Cryopreserved Platelets

To complement the phenotypic characterization of MPs, the functional contribution of cryopreserved platelets to hemostasis was evaluated using viscoelastic testing. EXTEM clotting time (Figure 3A) and clot formation time (Figure 3B) remained within clinical reference ranges, and no significant differences were observed between the control and DES groups, indicating preserved initiation and propagation of clot formation following thawing. Maximum clot firmness (MCF) was moderately reduced in the cryopreserved samples (Figure 3C), which is consistent with previous reports describing a partial impairment of clot stabilization after cryopreservation. Nevertheless, the observed MCF values provide in vitro evidence that DMSO-free platelets retain the capacity to contribute to coagulation, underscoring their potential functional relevance even in the setting of reduced firmness. Importantly, these findings suggest that the inclusion of DES may not be necessary when CRF is applied in DMSO-free cryopreservation, as no additional benefit was observed in clot quality or function.

## 3. Discussion

The present study evaluated whether the inclusion of a choline chloride-based deep eutectic solvent (DES) could enhance the efficacy of a recently developed DMSO-free NaCl cryopreservation protocol combined with controlled-rate freezing (CRF). Contrary to our initial hypothesis, the incorporation of DES did not yield measurable improvements across the evaluated parameters. Post-thaw platelet recovery, phenotype, mitochondrial integrity, microparticle profile, and functional contribution to hemostasis were all comparable between the DES-supplemented group and the NaCl/CRF control. These findings suggest that the physical optimization provided by CRF may already confer substantial protection, limiting the scope for additive-based improvements in this experimental setting. This interpretation is further substantiated by the data on uncontrolled freezing (Appendix A), which consistently demonstrated inferior outcomes under both isotonic and hypertonic NaCl conditions, thereby underscoring the protective role of CRF. Previous studies have reported that whole-blood results from TEG and ROTEM do not achieve acceptable limits of agreement and therefore cannot be considered interchangeable [17,18]. Such discrepancies, however, may be attributable to the use of different coagulation activators, which engage distinct hemostatic pathways. Notably, the MA findings under uncontrolled freezing conditions lend additional support to the conclusion that CRF provides a methodological advantage for optimizing the NaCl protocol.

How platelets withstand cryopreservation without conventional cryoprotectants likely reflects the combined effects of NaCl and CRF. As extracellular ice forms, rising NaCl concentrations drive osmotic dehydration, lowering intracellular water content and thereby reducing the risk of lethal ice formation. At the same time, gradual cooling allows this dehydration to keep pace with ice growth, preventing abrupt intracellular freezing events [19]. The resulting hypertonic environment may also stabilize platelet membranes by tightening lipid packing [20]. Finally, the small size and simple structure of platelets make them less prone to cracking than larger, nucleated cells. Together, these mechanisms may hypothetically explain how controlled freezing with NaCl alone achieves substantial cryoprotection.

It is well established that DMSO-cryopreserved platelets contain substantial amounts of platelet membrane vesicles (PMVs). Nevertheless, the mechanisms underlying platelet membrane alterations and the subsequent release of PMVs in cryopreserved platelets remain insufficiently understood, and the hemostatic properties of PMVs derived from cryopreserved platelet products (CPPs) have yet to be fully elucidated [21]. Our data indicate a relatively low concentration of MPs post-thaw, which aligns with the high recovery of intact platelets and a low increase in extracellular LDH. Collectively, these observations suggest that post-thaw cell destruction is limited under the conditions used in this study. Interestingly, our analysis revealed the presence of TLR2, TLR4, and GPVI on MPs, a finding not previously described in the literature; however, the data were highly inconsistent, and the underlying causes are not fully understood. One proposed explanation for this relates to selective encapsulation induced by freezing stress. Freezing stress during storage or processing may cause selective packaging of receptors into MPs, leading to variable expression levels of TLR2, TLR4, and GPVI on these MPs. This selective encapsulation could affect how these receptors functionally contribute to immune and hemostatic responses mediated by MPs. The significance of these findings for transfusion immunomodulation lies in how MPs bearing these receptors can modulate immune responses and platelet activity after transfusion. TLR2 and TLR4 are crucial in platelet activation and innate immune regulation, responding to bacterial components (e.g., lipopolysaccharides) and damage-associated molecular patterns, thus influencing inflammation and immune activation. GPVI is involved in platelet activation through collagen binding and plays a role in coagulation. Variability in their expression on MPs may impact the extent and nature of immune and thrombotic responses post-transfusion, potentially contributing to immunomodulation effects such as altered inflammation or thrombosis risk [22,23]; our results, therefore, emphasize the need for further systematic investigation in future studies.

Importantly, the overall impression is that the in vitro data presented—whether with or without DES supplementation—aligns with previously reported outcomes for platelet cryopreservation using DMSO [1,2,10]. This convergence suggests that controlled-rate freezing with NaCl alone can achieve functional and structural maintenance of platelets broadly comparable to the traditional DMSO-based approach.

The lack of observable benefit from DES inclusion should not be interpreted as a negative outcome. On the contrary, these results highlight the practical advantage of a simplified NaCl/CRF protocol for platelet cryopreservation. Eliminating both DMSO and additional cryoprotective agents reduces complexity, avoids potential cytotoxic effects associated with DMSO [24], and simplifies post-thaw processing. These practical considerations are especially relevant in transfusion medicine and emergency care, particularly in acute trauma, resource-limited hospitals, and military or prehospital environments, where logistical simplicity is paramount [25,26,27]. From a translational perspective, the demonstration that platelets can be cryopreserved effectively without either DMSO or alternative additives provides a strong rationale for further clinical evaluation.

Our results contrast with previous reports highlighting benefits of DES in cryobiology. For example, proline–glycerol eutectic formulations have been shown to support post-thaw recoveries and function comparable to DMSO-based protocols in cryopreserved platelets [16]. Likewise, DESs have demonstrated protective effects on protein stability and lipid membrane integrity in a variety of experimental systems [28]. A possible explanation for the discrepancy lies in the combined effects of protocol optimization and cell type specificity; while uncontrolled freezing in NaCl alone causes substantial platelet damage [10], CRF dramatically improves cryosurvival and may mask any incremental protective effect of DES. Thus, the advantage of DES may be context-dependent, emerging primarily under less-controlled freezing regimens or in cell types more vulnerable than platelets to cryo-induced stress.

Another important consideration is that our evaluation was restricted to a single DES formulation, which limits the generalizability of our findings. The physicochemical properties of DESs—particularly viscosity, polarity, and osmotic pressure—can significantly influence their interactions with biological membranes. High viscosity may hinder solute diffusion and impair penetration across the platelet membrane, potentially restricting intracellular protection during freezing and thawing. Conversely, the osmotic characteristics of the DES could alter membrane hydration states or induce transient stress responses, affecting membrane integrity and receptor functionality after cryopreservation. The observed ineffectiveness of the current DES may therefore reflect an unfavorable balance between these physicochemical factors and the structural resilience of the platelet membrane [11,12,13,14].

It remains possible that alternative or tailored DESs, optimized for specific cryoinjury pathways, such as ice recrystallization inhibition [9], oxidative stress mitigation [29], or cytoskeletal stabilization [30], could yield superior outcomes. Future work should explore a broader range of DES compositions and ratios to better modulate these interactions. Moreover, while our analysis incorporated a comprehensive set of in vitro functional and structural markers, in vivo validation is necessary to assess whether DES supplementation confers subtler benefits, including improved post-transfusion platelet circulation, clearance dynamics, or contributions to hemostatic efficacy in clinical contexts.

In summary, our data show that the choline chloride-based DES does not enhance in vitro platelet cryopreservation outcomes when used with the NaCl and controlled-rate freezing (CRF) protocol, affirming the robustness and simplicity of this DMSO-free approach. By preserving platelet phenotype and function without additional additives, the NaCl/CRF regimen reduces procedural complexity and minimizes toxicity risks, which has significant implications for streamlined emergency blood transfusions. Future research should focus on clinical validation of this regimen and systematically evaluate other hydrogen bond donor–acceptor (HBD–HBA) combinations in DES formulations. Such studies may reveal benefits under suboptimal freezing conditions, extended storage, or preservation of different blood cell types, thereby advancing DMSO-free cryopreservation technologies.

## 4. Material and Methods

### 4.1. Experimental Overview: Cryopreservation

Double-dose platelet concentrates (n = 10) were prepared from pools of eight ABO-identical buffy coats, suspended in SSP+ additive solution, following established protocols [10]. Each unit was divided into two equal parts and transferred into separate freezing bags (Macopharma, Mauvaux, France) using a sterile connection device (Terumo BCT, Tokyo, Japan) to maintain sterility. Twenty double-dose buffy coat-derived platelet (BC-platelet) units were used in total. Ten units were treated with a cryoprotective agent (Test group), while the remaining ten served as untreated controls.

A choline chloride–glycerol deep eutectic solvent (DES) was prepared by mixing choline chloride (ChCl, Sigma-Aldrich, Burlington, VT, USA) and glycerol at a 1:3 molar ratio. The resulting DES contained approximately 33.6% (*w*/*w*) ChCl, corresponding to a molar concentration of ~2.8 M based on its measured density (1.18 g·mL^−1^). For cryoprotectant formulation, the DES was diluted to 10% (*w*/*w*) in the final mixture, yielding an effective ChCl concentration of approximately 0.24 M. The DES units were pre-treated and stored on a flatbed agitator (60 cycles a minute, model LPR-3; Melco, Glendale, CA, USA) in a temperature-controlled cabinet, Helmer at 22 ± 2 °C for 20 min pre CRF. The optimal ChCl concentration was determined in platelet pilot studies, based on post-thaw platelet counts and JC-1 mitochondrial membrane potential staining.

All units (Test and Control) were then processed using a standardized NaCl protocol. Each unit received 100 mL of sterile 0.9% sodium chloride (NaCl), followed by gentle mixing. Afterward, the units were centrifuged at 1200× *g* for 10 min. The supernatant was carefully removed, leaving approximately 15 mL of concentrated platelet suspension per unit. This yielded a final residual concentration of approximately 0.16% DES/NaCl in the Test group and 0.3% NaCl in the Control group. Processed units were transferred to metal sheet storage boxes (Ninolab, Stockholm, Sweden) and cryopreserved using a controlled-rate freezing system (Kryo560, Planer, Sunbury-on-Thames, UK) at −80 °C. All units were stored at −80 °C for a minimum of 90 days. Cryopreservation at −80 °C for over 90 days is generally considered sufficient to minimize ice recrystallization and cell damage, as this temperature effectively halts cellular metabolism and slows molecular motion, while the extended storage period ensures complete cessation of biological processes. Thawing in AB plasma is a standard method for rehydrating and stabilizing cells without introducing cryoprotectant toxicity that could also cause artifacts.

For reconstitution, cryopreserved units were thawed and resuspended in 100 mL of freshly thawed AB plasma. Immediately post-thaw, platelet quality was assessed through a series of analyses evaluating cellular integrity, functional capacity, phenotypic markers, and early apoptosis. The study design is outlined in Figure 4. Platelets cryopreserved under uncontrolled freezing conditions in either isotonic or hypertonic NaCl, following a study design like that in Figure 4, are presented in Appendix A.

### 4.2. Analysis of Cell Count, Recovery, Disintegration and Blood Gas

Cellular in vitro parameters including Pree-freezing count, recovery post freezing of platelet counts, and mean platelet volume (MPV) were measured using the CA 620 Cellguard (Boule Medical, Stockholm, Sweden). To assess platelet disintegration post thaw, extracellular lactate dehydrogenase (LDH) activity was measured by spectrophotometry using an assay kit (Sigma-Aldrich N6660, St. Louis, MO, USA). Samples from the supernatant (centrifuged for 15 min at 3000 rpm) and platelet units (pre-treated 1:10 with 0.5% Triton) were prepared and frozen at −80 °C in phosphate buffer. Upon analysis, absorbance readings were recorded at 340 nm at 1 min intervals over 3 min using UV5Bio spectrophotometer (Mettler Toledo, Greifensee, Switzerland). LDH activity was calculated and expressed as a percentage of total available activity. We assessed the extracellular metabolic environment with routine blood gas analysis (ABL 800, Radiometer, Copenhagen, Denmark), measuring pH (37 °C), carbon dioxide and oxygen partial pressures (kPa at 37 °C), glucose, lactate, and bicarbonate. The measurements were carried out on the same plasma used for reconstitution, serving only as a quality check to ensure proper reconstitution. For this reason, the results are not discussed further.

### 4.3. Flow Cytometry Analysis

Flow cytometry was performed using a CytoFLEX Flow Cytometer (Beckman Coulter Life Sciences, Bromma, Sweden). Control specimens were processed and incubated with a FITC- or PE-conjugated mAb (IgG1 isotype) with irrelevant specificity, purchased from Immunotech (Beckman Coulter). Platelet samples, fixed by adding an equal volume of 1% paraformaldehyde (PFA), PFA-phosphate-buffered saline (pH 7.2–7.4) at 22 °C for 10 min, were then stained for 20 min at the same temperature in the dark by incubating with 20 µL of fluorochrome-labeled monoclonal antibodies (mAb) per 1 × 10^6^ platelets. The following markers were analyzed above that of background (control): P-selectin (CD62P): PE-conjugated (Beckman Coulter), (CD63): FITC-conjugated (Beckman Coulter) GPIb (CD42b): FITC-conjugated (Beckman Coulter), GPIIb (CD61): PE-conjugated (Beckman Coulter), (CD41a): FITC-conjugated (Beckman Coulter), GPVI: PE-conjugated (Pharmingen BD Biosciences), PECAM-1 (CD31): (Sigma-Aldrich). The conformational activation of the GPIIb/IIIa complex was assessed using FITC-conjugated PAC-1 monoclonal antibody (IgM, Becton Dickinson, Franklin Lakes, NJ, USA). TLR2 and TLR4 (Toll-like receptor 2 and 4): PE-conjugated (Pharmingen BD Biosciences, San Jose, CA, USA) and lysosomal-associated membrane protein (LAMP-1): (R&D systems, Minneapolis, MN, USA). Mitochondrial membrane potential (Δψm) was evaluated using the MitoPT JC-1 detection kit (Immuno-Chemistry Technologies, Bloomington, MN, USA) as recently described [10]. Microparticles (MPs) were analyzed as a percentage of the total extracellular vesicle (EV) population by flow cytometry, using size-calibrated beads (Biocytex, Marseille, France) to accurately gate and discriminate MPs within the EV fraction. Subpopulations of MPs were identified using the surface markers CD61, TLR2, TLR4, and GPVI.

### 4.4. Thromboelastometry

Hemostatic function was assessed using thromboelastometry (ROTEM delta 3000, TEM International, Munich, Germany). Cryopreserved DES/NaCl-treated and control (NaCl-only) platelets were diluted in fresh frozen AB plasma to a concentration of 300 × 10^9^ platelets/L. EXTEM assays were performed per the manufacturer’s instructions to evaluate clot formation dynamics. Maximum clot firmness (MCF), which reflects the maximum tensile strength of the thrombus, clot formation time (CFT), namely the time that clot takes to increase from 2 to 20 mm above baseline, and clotting time (CT) were recorded post thawing within 2 h, with all assays performed at a controlled temperature of 37 °C to maintain physiological conditions. For the uncontrolled freezing analysis (Appendix A), viscoelastic hemostatic testing was performed using the TEG^®^ 6s Hemostasis Analyzer System (Haemonetics Corporation, Braintree, MA, USA). PlateletMapping^®^ ADP and AA Assay Cartridges, preloaded with channel-specific reagents, were employed to evaluate clot formation mediated by the adenosine diphosphate (ADP) and arachidonic acid (AA) pathways. The erythrocyte volume fraction (EVF) was standardized at 30% across all assays. For each measurement, 340 μL of platelet suspension (30% EVF) was dispensed into the designated cartridge chambers following the manufacturer’s protocol. All analyses were completed within two hours of sample collection, with the temperature of both samples and instrumentation maintained at 37 °C throughout testing to ensure consistency and comparability. The TEG 6s system reported maximal amplitude (MA) values for both ADP and AA channels, enabling direct comparison of their respective contributions to clot strength. Measurements were performed in duplicate, with quality control procedures and instrument calibration carried out per manufacturer recommendations to ensure analytical consistencies.

### 4.5. Statistics

All statistical analyses were performed using GraphPad Prism (version 10.2.2, build 397). Data are presented as mean ± standard deviation (SD) along with 95% confidence intervals (CI). Comparisons between paired measurements were conducted using paired Student’s t-test. Statistical significance was set at *p* < 0.05. The assumption of normality was evaluated on the residuals using the Anderson–Darling, D’Agostino–Pearson omnibus, Shapiro–Wilk, and Kolmogorov–Smirnov tests; all datasets met normality criteria.

## Figures and Tables

**Figure 1 ijms-26-10013-f001:**
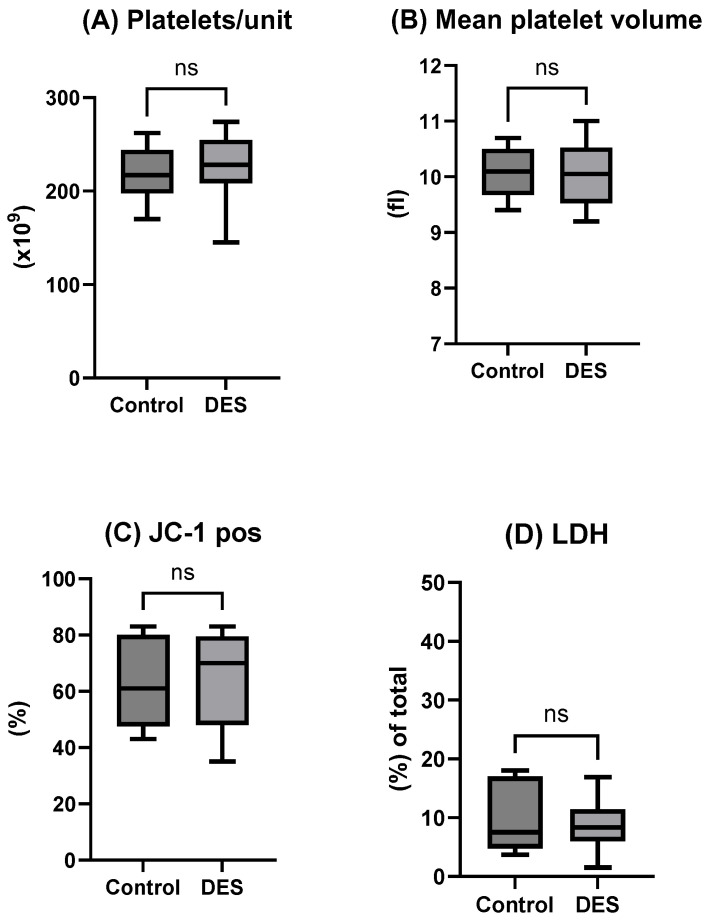
Data are presented as mean ± SD (n = 10) and include platelet count (**A**), mean platelet volume (**B**), JC-1^+^ (**C**), and LDH, % of total (**D**) post-thaw. Statistical significance of differences was defined as *p* < 0.05 for comparisons between control versus deep eutectic solvent (DES)-treated samples subjected to controlled-rate freezing (CRF) cryopreservation. ns: not statistically significantly different.

**Figure 2 ijms-26-10013-f002:**
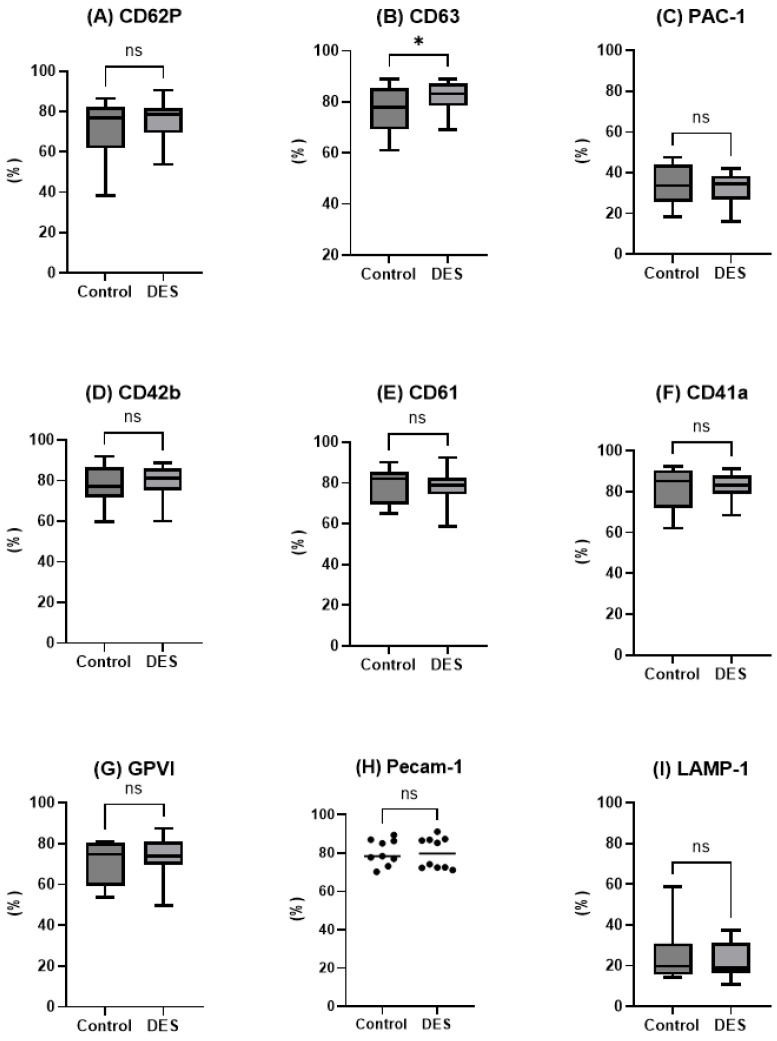
Expression of platelet activation, adhesion, signaling, and degranulation markers CD62P (**A**), CD63 (**B**), PAC-1 (**C**), CD42b (**D**), CD61 (**E**), CD41 (**F**), GPVI (**G**), PECAM-1 (**H**), LAMP-1 (**I**) on DMSO-free cryopreserved platelets (control) and on platelets supplemented with choline chloride (deep eutectic solvent, DES). Data represent the percentage of positive cells (mean ± SD). * *p* < 0.05.

**Figure 3 ijms-26-10013-f003:**
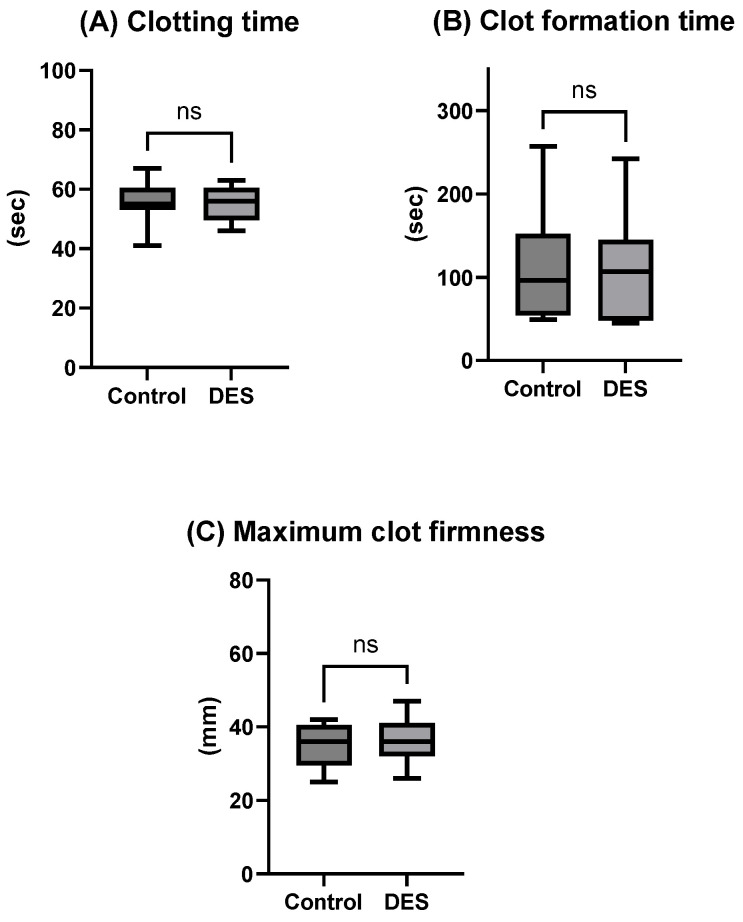
Viscoelastic properties: EXTEM clotting time (**A**), clot formation time (**B**) and maximum clot firmness (**C**) assessed on the cryopreserved platelets post-thaw. Data is presented as mean ± SD of n = 10. Results are presented post-1–2 h after thawing for the DMSO-free CRF-cryopreserved platelets (control) compared to DMSO-free cryopreserved platelets supplemented with choline chloride (deep eutectic solvent, DES) platelets, respectively. ns: not statistically significantly different.

**Figure 4 ijms-26-10013-f004:**
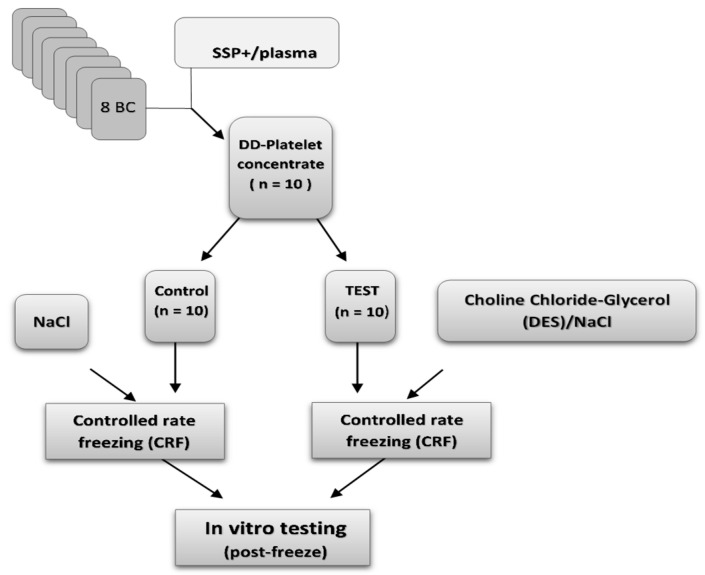
Schematic overview of the paired two-armed study design.

**Table 1 ijms-26-10013-t001:** Microparticle (MP) phenotypes as percentages of total extracellular vesicles (EVs), with no significant differences observed between groups for the surface markers CD61, TLR2, TLR4, and GPVI. Size-calibrated beads (Megamix-Plus SSC, Biocytex, Marseille, France) were used to accurately gate and discriminate MPs within the EV fraction. These fluorescent polystyrene beads, covering a range of diameters from approximately 160 nm to 500 nm, provided a reliable size reference for defining the microparticle gate based on side scatter characteristics. This allowed precise discrimination of MPs from other EV subpopulations and background noise, ensuring reproducibility across measurements. Subpopulations of MPs were further identified using the surface markers CD61, TLR2, TLR4, and GPVI, enabling detailed phenotypic characterization of platelet-derived and immune-related MPs. All results shown as mean ± standard deviation (SD).

Microparticles (MPs) % of EV Total	MV ± SD (Control)	MV ± SD (DES)
CD61 pos MPs	41 ± 11	46 ± 10
TLR2 pos MPs	15 ± 15	16 ± 16
TLR4 pos MPs	18 ± 18	14 ± 13
GPVI pos MPs	13 ± 9	14 ± 11
Phenotypic expression (%)	79 ± 10	81 ± 6
CD61	79 ± 9	77 ± 9
TLR2	33 ± 19	32 ± 18
TLR4	33 ± 19	34 ± 19
GPVI	72 ± 10	73 ± 11

## Data Availability

Raw data are currently unavailable due to Karolinska policy restrictions.

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
