# Peer review of "Innovations in Platelet Cryopreservation: Evaluation of DMSO-Free Controlled-Rate Freezing and the Role of a Deep Eutectic Solvent as an Additional Cryoprotective Agent"

_ijms, 2025, doi:10.3390/ijms262010013_

Round 1
Reviewer 1 Report
Comments and Suggestions for Authors
The study presents a comprehensive investigation into DMSO-free controlled-rate freezing for platelet cryopreservation and the potential additive role of a choline chloride-glycerol deep eutectic solvent as a cryoprotectant. The manuscript is well-structured, and the combination of cellular integrity assessments, phenotypic profiling, and functional hemostatic analyses provides valuable insights into the feasibility of simplified NaCl-based protocols. However, several points require further revision:
1 The manuscript states that DES exposure was applied for 20 minutes prior to NaCl processing, but the methodology does not explicitly describe the rationale for this specific ratio, concentration, or incubation time. Please clarify these selections, including any preliminary optimization data, to enhance reproducibility.
2 Table 1 presents microparticle phenotypes as percentages of total extracellular vesicles, with values for CD61, TLR2, TLR4, and GPVI showing no significant differences between groups. The text does not specify the gating strategy or bead calibration details beyond a brief mention; please add these to confirm methodological rigor.
3 All cryopreservation was conducted at -80°C for over 90 days, with thawing in AB plasma. A brief justification for the storage duration and temperature should be added to confirm they do not introduce artifacts, such as potential ice recrystallization, that could confound the comparison between DES-treated and control units.
4 Some figures are dense; please simplify by grouping related markers or adding insets for clarity.
Reviewer 2 Report
Comments and Suggestions for Authors
The authors evaluated the protective effect of adding a choline chloride-glycerol deep eutectic solvent to a DMSO-free platelet cryopreservation protocol (using controlled-rate freezing and saline). In vitro comparisons revealed that the addition of DES did not significantly improve platelet recovery, phenotype, function, or coagulation capacity compared to a saline-only control. This study demonstrates that a simplified protocol combining controlled-rate freezing with saline effectively maintains platelet quality without the need for additional DES, providing important insights for developing safer and more convenient platelet cryopreservation strategies. This article is acceptable for publication with minor revisions.
1. The statement "No significant differences were observed" in the Abstract is somewhat vague. It is recommended that the specific numerical ranges for key indicators be clearly stated to enhance the persuasiveness. Keywords should also be supplemented with terms such as "DMSO-free cryopreservation" to improve search relevance.
2. The introduction could be more concise in its description of the potential of DES, and the second paragraph describing the general characteristics of DES should be streamlined. The differences and innovations of this study compared to existing DES studies (e.g., the Pro-Gly system) need to be more clearly highlighted.
3. The description of DES treatment conditions in the study methods is incomplete; key information such as treatment temperature and specific agitation parameters should be supplemented. The experimental basis for selecting a 10% DES concentration should also be explained to strengthen the rationale of the protocol.
4. The analysis of the reasons for DES ineffectiveness in the Discussion section is insufficiently detailed; the interaction between DES physicochemical properties (e.g., viscosity and osmotic pressure) and the platelet membrane should be considered. Limitations of this study, such as the single DES formulation, need to be more clearly highlighted.
5. A more specific mechanistic hypothesis should be proposed for the inconsistent expression of TLR2/TLR4/GPVI in MPs, such as selective encapsulation induced by freezing stress. A brief discussion of the significance of these findings for transfusion immunomodulation is recommended.
6. The flow cytometry section in the Materials and Methods section should include additional information on key parameters such as antibody incubation time and temperature to ensure reproducibility. Thromboelastography testing should clearly state the time window and temperature control conditions for sample testing.
7. The conclusion section should highlight the clinical value of the CRF/NaCl regimen, particularly the significance of simplified procedures for emergency blood transfusions. Recommendations for future research should be specific, such as recommending testing other HBD-HBA combinations in DES formulations.
Comments on the Quality of English LanguageThe authors evaluated the protective effect of adding a choline chloride-glycerol deep eutectic solvent to a DMSO-free platelet cryopreservation protocol (using controlled-rate freezing and saline). In vitro comparisons revealed that the addition of DES did not significantly improve platelet recovery, phenotype, function, or coagulation capacity compared to a saline-only control. This study demonstrates that a simplified protocol combining controlled-rate freezing with saline effectively maintains platelet quality without the need for additional DES, providing important insights for developing safer and more convenient platelet cryopreservation strategies. This article is acceptable for publication with minor revisions.
1. The statement "No significant differences were observed" in the Abstract is somewhat vague. It is recommended that the specific numerical ranges for key indicators be clearly stated to enhance the persuasiveness. Keywords should also be supplemented with terms such as "DMSO-free cryopreservation" to improve search relevance.
2. The introduction could be more concise in its description of the potential of DES, and the second paragraph describing the general characteristics of DES should be streamlined. The differences and innovations of this study compared to existing DES studies (e.g., the Pro-Gly system) need to be more clearly highlighted.
3. The description of DES treatment conditions in the study methods is incomplete; key information such as treatment temperature and specific agitation parameters should be supplemented. The experimental basis for selecting a 10% DES concentration should also be explained to strengthen the rationale of the protocol.
4. The analysis of the reasons for DES ineffectiveness in the Discussion section is insufficiently detailed; the interaction between DES physicochemical properties (e.g., viscosity and osmotic pressure) and the platelet membrane should be considered. Limitations of this study, such as the single DES formulation, need to be more clearly highlighted.
5. A more specific mechanistic hypothesis should be proposed for the inconsistent expression of TLR2/TLR4/GPVI in MPs, such as selective encapsulation induced by freezing stress. A brief discussion of the significance of these findings for transfusion immunomodulation is recommended.
6. The flow cytometry section in the Materials and Methods section should include additional information on key parameters such as antibody incubation time and temperature to ensure reproducibility. Thromboelastography testing should clearly state the time window and temperature control conditions for sample testing.
7. The conclusion section should highlight the clinical value of the CRF/NaCl regimen, particularly the significance of simplified procedures for emergency blood transfusions. Recommendations for future research should be specific, such as recommending testing other HBD-HBA combinations in DES formulations.
